# Personal Genomes in Practice: Exploring Citizen and Healthcare Professionals’ Perspectives on Personalized Genomic Medicine and Personal Health Data Spaces Using a Mixed-Methods Design

**DOI:** 10.3390/genes14040786

**Published:** 2023-03-24

**Authors:** Judith Tommel, Daan Kenis, Nathalie Lambrechts, Richard M. Brohet, Jordy Swysen, Lotte Mollen, Marie-José F. Hoefmans, Murih Pusparum, Andrea W. M. Evers, Gökhan Ertaylan, Marco Roos, Kristien Hens, Elisa J. F. Houwink

**Affiliations:** 1Health, Medical, and Neuropsychology Unit, Institute of Psychology, Faculty of Social and Behavioural Sciences, Leiden University, Wassenaarseweg 52, 2333 AK Leiden, The Netherlands; 2Department of Psychiatry, University of Groningen, University Medical Centre Groningen, Hanzeplein 1, 9713 GZ Groningen, The Netherlands; 3Department of Philosophy, University of Antwerp, Rodestraat 14, 2000 Antwerp, Belgium; 4VITO Health, Boeretang 200, 2400 Mol, Belgium; 5Department of Epidemiology and Statistics, Isala Hospital, Dokter van Heesweg 2, 8025 AB Zwolle, The Netherlands; 6Schluss, Overhoeksplein 2, 1031 KS Amsterdam, The Netherlands; 7Data Science Institute, I-Biostat, Hasselt University, 3500 Hasselt, Belgium; 8Department of Human Genetics, Leiden University Medical Center, Albinusdreef 2, 2333 ZA Leiden, The Netherlands; 9Department of Family Medicine, Mayo Clinic, 221 Fourth Avenue SW, Rochester, MN 55905, USA; 10Department of Public Health and Primary Care (PHEG), Leiden University Medical Center, Albinusdreef 2, 2333 ZA Leiden, The Netherlands

**Keywords:** genomic medicine, personalized medicine, mixed-methods, personal health data space, ethics, genetic data, Belgium, The Netherlands

## Abstract

Ongoing health challenges, such as the increased global burden of chronic disease, are increasingly answered by calls for personalized approaches to healthcare. Genomic medicine, a vital component of these personalization strategies, is applied in risk assessment, prevention, prognostication, and therapeutic targeting. However, several practical, ethical, and technological challenges remain. Across Europe, Personal Health Data Space (PHDS) projects are under development aiming to establish patient-centered, interoperable data ecosystems balancing data access, control, and use for individual citizens to complement the research and commercial focus of the European Health Data Space provisions. The current study explores healthcare users’ and health care professionals’ perspectives on personalized genomic medicine and PHDS solutions, in casu the Personal Genetic Locker (PGL). A mixed-methods design was used, including surveys, interviews, and focus groups. Several meta-themes were generated from the data: (i) participants were interested in genomic information; (ii) participants valued data control, robust infrastructure, and sharing data with non-commercial stakeholders; (iii) autonomy was a central concern for all participants; (iv) institutional and interpersonal trust were highly significant for genomic medicine; and (v) participants encouraged the implementation of PHDSs since PHDSs were thought to promote the use of genomic data and enhance patients’ control over their data. To conclude, we formulated several facilitators to implement genomic medicine in healthcare based on the perspectives of a diverse set of stakeholders.

## 1. Introduction

The current study aimed to assess attitudes towards *genomic data* and *personal health data management tools* among healthy participants, healthcare users, and professionals as a means to identify key concerns and interests regarding *personalized medicine*. Personalized or precision medicine encompasses the application of a variety of biomedical and technological advancements in clinical research and healthcare, positioned to address novel and persistent challenges to existing healthcare systems [1]. The proposed strategies for tackling issues, such as the increased global burden of chronic disease, primarily involve optimizing and personalizing healthcare by attending to individuals’ unique clinical, genetic, genomic, and environmental information to guide medical decision-making [2]. At the same time, personalized medicine is committed to reshaping the contemporary clinical landscape by placing patients at the center of their care trajectory, turning them from “mere passengers” to “active drivers” [3]. It is then suggested that patient empowerment is achieved by providing access to health information and enabling participatory clinical decision-making with the healthcare provider (HCP) [4].

Genomic medicine is a crucial component of contemporary approaches to personalized healthcare. Genomic medicine applies the tools of genomic analysis, notably whole genome sequencing (WGS), to detect subgroups of patients and to assign an individual patient to a specific subset of patients that responds particularly well to a particular intervention (i.e., stratification). In this capacity, genomic medicine can be applied to risk assessment, disease prevention, (early) diagnosis, prognosis, therapeutic selection, and monitoring (recurrence) [2,5]. In cancer treatment, for example, genomic medicine enables earlier detection and better prognostic prediction [5]. Using therapeutic selection, clinicians aim to decrease side effects by selecting treatment that is known to exhibit fewer side effects in a particular subgroup with a specific genetic variation (i.e., pharmacogenomics) [1,5]. 

Despite these favorable prospects, several challenges must be addressed to ensure the successful, equitable, and desirable implementation of genomic medicine in primary care. These challenges can be divided into three (interlocking) areas of concern: practical, ethical, and technological. Starting with practical challenges, healthcare providers have reported a lack of knowledge and time, limited access to genomic expertise and testing, lack of clinical and ethical standards for genomic application, difficulties in integrating genomic results into electronic health records (EHRs), and a lack of clinical decision support [5,6,7,8,9,10]. To overcome these barriers, three essential strategies have been identified to improve the adoption of genomic medicine: (i) standardized and assessable collection and analysis methods; (ii) healthcare provider access to genomic data; and (iii) clinical guidance for interpretation and use [2]. 

Regarding ethical challenges, previous studies suggest that the public (e.g., healthy volunteers, patients, healthcare providers) tends to acknowledges the potential benefits of genetic research and genomic medicine [7,11,12,13]. However, the inclusion of genomics in primary care can profoundly impact patients and their families and raises several ethical, legal, and social issues (ELSIs). These concerns include consent, disclosure of individual findings, data sharing, privacy, and confidentiality [14]. A qualitative study focused on women with a personal or family history of breast cancer found, for example, that these women recognized the possibilities of genomics to improve their own or their family’s health. Still, they worried about losing control over their health, privacy, and confidentially and raised questions about the accuracy and uncertainty of genetic testing [12]. Other worries involve racial and/or genetic discrimination (e.g., in insurance or employment), short-term distress caused by incidental or secondary findings, and increased healthcare risks and costs due to additional clinical follow-up indicating the need for robust, trustworthy institutions and other support services, such as genetic counseling [5,10,11,15,16,17]. Critics argue that the current operationalization of empowerment in healthcare, such as centralizing patient involvement in care and providing health information, may lead to increased individual responsibility for health, calling for more patient-driven conceptions of empowerment [18,19,20,21]. These studies and commentaries demonstrate the need to gain additional insight into population attitudes toward genomic and personalized medicine to ensure proper implementation in existing healthcare systems.

Regarding technological challenges, robust clinical structures, data infrastructure, and standards are currently lacking [5,11,13]. This is highly problematic, especially since digital platforms that incorporate proper data infrastructure and clinical standards help to efficiently collect and register genomic data in a standardized format, making it easier to incorporate genomic information into EHRs and support evidence-based decision-making according to current clinical genetic guidelines [2,8]. In this way, digital platforms facilitate the realization of genomic medicine by supporting clinicians in an efficient and time-sensitive manner [8]. Moreover, the Belgian Health Care Knowledge Center (KCE) suggests the development of secure, accessible, yet transparent data infrastructure as a significant hurdle for implementing WGS in clinical care [22]. These concerns are also reflected in the European Union’s (EU) proposal for the European Health Data Space (EHDS) as a key pillar of its health (data) strategy [22]. The EHDS aims to establish a reliable, interoperable, and secure environment for health data for all EU citizens [23]. Nominally, the EHDS proposal envisions a cross-border health data ecosystem that bridges the gap between research, care contexts, and individual EU citizens. Moreover, by providing rules, standards, infrastructure, and governance frameworks, the EHDS proposal seeks to unlock the potential of an EU-wide health data infrastructure to benefit all actors involved. Despite these commendable resolutions, several scholars, regulatory bodies (including the European Data Protection Board), and advocacy groups (such as the European Patients Forum) have expressed worries regarding the current proposal [24,25,26,27]. Their main concern is the lack of engagement with citizens and patients, especially in the secondary use of health data. The EHDS proposal leaves sufficient leeway to maintain or reinstate data asymmetries between the public and private sectors, running the risk of falling short of its laudable vision for an equitable, transparent, and ultimately empowering health data ecosystem. 

To address these issues, patient-oriented Personal Health Data Spaces (PHDSs) are being developed. Early European projects, such as Decode, MyHealthMyData, MIDATA, and MedMij, however, have been limited in scope and do not ensure sufficient health data sharing across industry, institutions, and citizens [28,29]. What is necessary, then, are dynamic, interoperable data ecosystems balancing data access, control, and use tailored to diverse stakeholder needs. 

Several of these next-generation data management platforms are currently under development. The Personal Genetic Locker project in the Netherlands aims to design a tool that not only provides clinicians with necessary genetic knowledge and support but also has the potential to enhance the individual’s sense of control over their health and their data [30,31]. The personal genetic locker (PGL) aims to provide a safe, user-friendly, interoperable digital environment where individuals can store, view, and interpret their genomic information [31]. Additionally, it allows individuals to consent to or prevent sharing data with healthcare providers or researchers (Figure 1). Combined with analytic tools, including a clinical decision support system based on up-to-date clinical guidelines, the PGL can be integrated in every EHR to facilitate the adoption of genomic medicine in clinical practice [30,31]. Other notable next-generation projects include TiDaL (Netherlands [29]) and We Are (Belgium [32]). By establishing so-called personal data vaults or pods (following FAIR principles), personal health data are stored in a decentralized manner and “visited” rather than moved upon analysis (e.g., by federated data analysis), ensuring anonymity and privacy by separating data from application [29,31,33]. These next-generation health data management platforms are often positioned as intermediaries between the EHDS provisions (focused on cross-border research and the commercial reuse of health data) and individual citizens. By positioning themselves as explicitly person-centered and by providing citizens with increased data control, access, and transparency, PHDS initiatives aim to enhance individual agency over data management in accordance with the principles for secondary use envisioned in the EHDS proposal. 

To ensure the implementation of genomic medicine, for PHDS initiatives, such as the PGL, to match the needs and wishes of future users and given the ethical concerns raised in the literature, the current study aimed to evaluate healthy participants’, healthcare users’, and healthcare professionals’ perspectives on genomic analysis and PHDS solutions with the PGL as an example. Using focus groups, interviews, and a questionnaire survey, we assessed attitudes towards genomic analysis and PHDSs, including possible (practical or ethical) concerns, potential benefits, required functionalities, and barriers to and facilitators of implementing genomic medicine in clinical care. Additionally, by involving future users in an early stage of the development process, our study contributes to (i) a genomic healthcare organization and a PHDS design that is user-friendly and informed by the needs of both healthcare users and professionals, (ii) facilitating the implementation of genomic medicine and PHDSs in clinical care, and (iii) facilitating the secondary use of personal health data controlled by the individual. 

Two disclaimers are in order. In the current study, and in accordance with the scope of the PGL project, we focus specifically on genomic data. While many other data types (proteomics, epigenomics, metabolomics, etc.) will potentially be included in personal health data vaults, they might pose unique challenges (e.g., health literacy, reversibility, identifiability) that must be addressed thoroughly in their own right [34,35]. Second, researchers’ needs and concerns for the secondary usage of health data differ significantly from those of citizens. These concerns are, however, beyond the scope of the current study as we focus instead on the needs of HCPs, citizens, and healthcare users. 

## 2. Methods

### 2.1. Design

To gain a deeper, nuanced understanding of the attitudes of healthcare users and professionals towards genomic medicine, we applied a mixed-methods design that included a questionnaire survey, interviews, and focus groups in a collaborative effort among the University of Antwerp (Belgium), VITO (Belgium), Leiden University, and the Leiden University Medical Centre (The Netherlands).

For Belgium, the survey and interview study were carried out as part of VITO’s IAM Frontier study, a longitudinal precision medicine feasibility study aiming to define individual health parameters and investigate the implementation of personalized health profiles for personalized preventive medicine in clinical practice. All 30 individuals in the IAM Frontier study had their genomes sequenced via WGS. Individual sequencing results were returned through the study doctor following the ACMG guidelines for reporting genome analysis results [36]. If additional information was requested, participants were referred to the Center for Medical Genetics of the University Hospital of Antwerp (UZA). IAM participants were asked to complete an online survey before and after receiving individual personal genome analysis results. Six participants were invited for in-depth follow-up interviews after each survey (Figure 2). The design of the study was verified by a clinical geneticist.

The focus groups were conducted as part of the Dutch Personal Genetic Locker project that aims to define standards and prototypes for a safe environment where individuals can store, view, and interpret their personal genetic data in a personal health data space. For an elaborate description of the study aims and design, see Overkleeft et al. [31]. Two focus group discussions were organized: one with healthcare users and one with healthcare professionals [31]. 

The primary aims of the IAM Frontier and PGL projects—i.e., gathering perspectives on WGS and genomic data management through qualitative methodologies—show sufficient overlap to allow for a meaningful integration of the results. Importantly, each project offers unique and complementary insights strengthening the overall analysis. As such, we opted for a mixed-method design to gain a more comprehensive and nuanced understanding of attitudes, barriers, and facilitators in implementing genomic or personalized medicine. Mixed-method research combines quantified data and participant experience using quantitative and qualitative methodologies [37]. This study combines in-depth interviews, focus groups, and quantitative surveys of both projects. 

### 2.2. Participants

#### 2.2.1. IAM Frontier Surveys 

Participation in IAM Frontier was limited to VITO employees. Prospective participants, all working in a research context, were contacted through a company-wide call, internal communication, and information sessions on the project. We strived for gender diversity within the sample. Other inclusion criteria consisted of the following: being over 45 in age, being healthy (in casu, not suffering from a previously diagnosed chronic disease), being an active smartphone user, and having research-oriented motivation to participate. Frequent blood donors and chronic disease patients (asthma, COPD, chronic bronchitis, emphysema, myocardial infarction, coronary heart disease, other severe heart diseases, stroke, diabetes, and cancer (malignancies including leukemia and lymphoma)) were not eligible for participation. In light of potential conflicts of interest, members of the Health Department were excluded. The recruitment was carried out in January and February 2019. 

#### 2.2.2. IAM Frontier Interviews

Participants for the in-person interviews were selected from the initial IAM Frontier study population based on their answers on the first online survey. At the end of the online survey, participants were given the option to choose whether or not they wanted to participate in a semi-structured in-depth, in-person interview. Participants who agreed to this were deemed eligible for the interviews. The participants for these in-person interviews were selected from the 30 IAM Frontier study participants based on their answers to the first online survey. All 30 individuals filled in the survey, and 28 were on time to be selected for the interviews. Of this initial sample, 21 individuals wanted to participate in the in-depth interviews. From this sample, participants were chosen based on the following criteria: (1) high prevalence of the answer “yes” or “no” on three-point nominal “yes-no-don’t-know” scale questions and (2) high prevalence of answers on the extremes of the 5-point Likert-scale questions (selection procedure illustrated in Figure 2). Additionally, we aimed for diversity across gender and within age categories according to the inclusion criteria. Interviewees were recruited in January 2020 (see Figure 3).

#### 2.2.3. PGL Focus Groups 

Potential participants were recruited using purposive sampling in the Netherlands. The healthcare users’ focus group participants were recruited via the Dutch patient association on rare and genetic diseases (VSOP), online advertisements, and the project members’ network. As advised by the focus group literature (for example, Powell & Single, 1996 [38]), focus groups should comprise participants from a diverse range of backgrounds, views, and experiences in order to identify important issues and phenomena. Therefore, the aim was to recruit a diverse group of individuals, including participants familiar with genetic testing or genetic data and participants with little or no experience with genetic testing or data. Participants in the healthcare professionals focus group were recruited using the professional networks of the project members. The aim was to recruit a diverse group of healthcare professionals, such as physicians, specialists, and pharmacists. Exclusion criteria for both focus groups were having an age < 18 years, not being fluent in Dutch, and being a member of the PGL project. Additionally, for inclusion in the healthcare professionals focus group, participants should work as healthcare professionals at a healthcare facility in the Netherlands. Recruitment took place from April 2021 through June 2021. 

### 2.3. Procedures 

#### 2.3.1. IAM Frontier Surveys

Participation in the IAM Frontier study was voluntary. All participants gave their informed consent together with the researcher to ensure that they understood all aspects of the study and had no further questions before enrollment.

Participants were contacted by email to fill out the online survey. Surveys were answered by the participants twice during the IAM Frontier study (February and July–August 2020). Before feedback on individual whole genome sequencing results, all participants completed the first online survey covering the following topics: knowledge and experience with WGS (5-point Likert type scale anchored by “strongly disagree” to “strongly agree”), attitudes towards receiving potential WGS findings (three-point nominal “Yes-No-Don’t know” scale), ownership of genomic data (donor, research institute, or healthcare professional), sharing results with blood relatives, and social values of WGS (5-point Likert type scale; hereafter referred to as “pre”). 

After feedback on individual genome results (July 2020), participants were asked to complete the second online survey (hereafter referred to as “post”). This survey asked similar questions, including additional questions on the impact of individual genome results on participants’ lives (a three-point nominal “Yes-No-Don’t know” scale). The online surveys were developed based on existing literature and refined to fit the research questions. Online surveys were implemented in a VITO-developed and hosted online survey tool that guarantees privacy and the anonymous processing of survey data.

#### 2.3.2. IAM Frontier Interviews 

In-depth semi-structured interviews were set up to provide additional, more nuanced insight into the survey results. Interviews were conducted between February 2020 and March 2020 (hereafter referred to as “pre”), prior to receiving the individual WGS results, and in August 2020, approximately one month after communication of the individual WGS results (hereafter referred to as “post”). Interviews lasted about 50 min. Participant data were pseudonymized. Interview guides were developed based on existing literature and aligned with survey topics.

#### 2.3.3. PGL Focus Groups

All potential participants received information regarding the study procedures, emphasizing confidentiality. Participation was voluntary, and all participants signed an informed consent form prior to participation. 

The focus group discussions were conducted in July 2021 and were held online. An interview protocol was used to guide the discussions. The interview protocol was developed in collaboration with several members of the PGL project and followed guidelines on qualitative and focus group research (Appendix A) [39,40,41]. The focus group discussions were monitored by an independent moderator (JT). An observer took field notes on group dynamics and nonverbal communication and helped participants with minor technical problems during the session (e.g., logging in). The sessions took approximately 2.5 h and were audio-recorded.

The sessions started with an introduction to PDSs, including a demonstration of a mockup PGL designed by Schluss (foundation that develops personal data solutions) and a case study to illustrate the possible applicability of PGLs. Subsequently, participants were asked for their opinions regarding genetic data and a PGL (e.g., participants’ thoughts, needs, wishes, and concerns). Finally, the required functionalities and characteristics of a PGL were discussed. 

### 2.4. Data Analysis

#### 2.4.1. IAM Frontier Surveys

Frequency tables were used to report survey responses. No additional statistical analyses were conducted because this did not fall within the scope of the (pilot) study and the small sample size did not allow for sufficiently powered analyses. 

#### 2.4.2. IAM Frontier Interviews

Interviews were audio-recorded and transcribed verbatim. Data for qualitative interviews were presented and interpreted following the standards for reporting qualitative research [42]. Qualitative data were analyzed using thematic analysis (TA) [43]. TA refers to a range of diverse modes of qualitative data analyses, differing in their commitment to (post) positivist qualitative research values [44]. For IAM Frontier, a *reflexive* approach to TA was followed. Reflexive TA is distinguished from positivist approaches to qualitative research and conceives of coding practices as inherently interpretative and subjective. Reflexive TA embraces researcher reflexivity as an analytical tool [44]. After data familiarization, DK, a junior researcher of the team, created *inductive* codes, among which patterns were identified and themes were generated. In reflexive TA, themes are understood as capturing a core idea or meaning [44]. While not usually part of reflexive TA [44], KH, an experienced member of the research team reviewed the coding scheme and themes sparking additional reflexivity and enriching the initial analysis. Data were coded and managed using the NVivo PRO 12 software. 

#### 2.4.3. PGL Focus Groups

The focus groups were audio-recorded and professionally transcribed verbatim. We analyzed the transcripts using a coding reliability approach to thematic analysis [45,46,47]. Two independent coders analyzed the data to promote interpretation consistency and satisfactory inter-rater reliability. Using an inductive approach, JT and EJFH independently assigned codes and themes to the data. Following the coding reliability approach, themes are understood as summaries of topics [44]. Subsequently, JT and EJFH compared and discussed their results and decided upon a final codebook describing a definitive list of codes and themes, a coding label and definition of each code and theme. Subsequently, data coding was repeated using the revised codes and themes as described in the codebook. ATLAS.ti 9 was used for technical support. 

#### 2.4.4. Triangulation

Across the different methodologies and datasets from both study designs, we generated “meta-themes” [31] that provide additional insight into the respective datasets and allow meaningful integration of results. Importantly, triangulation was not applied for confirmatory purposes, but rather to offer a richer analysis informed both by a broader range of stakeholder perspectives as well as a confrontation between individual researcher’s situated perspectives—bringing this study closer to the distinctively postpositivist aims of reflexive TA as outlined by Braun and Clarke [44].

Three triangulation strategies were followed. First, multiple researchers engaged in the triangulation procedure—both in an individual capacity, as well as in discussions related to the results of triangulation. Second, different methodologies were triangulated. Third, multiple stakeholders in the respective datasets allowed for triangulation across these perspectives. 

A triangulation protocol adapted from Farmer et al. [48] was utilized and fit to the research design. After individual code and theme generation for each respective (qualitative) dataset (Section 2.4.2 and Section 2.4.3), findings were compared in relation to the meaning of the respective themes. Both convergences and dissonances between respective datasets and themes were noted, as well as topics only addressed in specific datasets. After individual assessments, JT (health psychologist) and DK (pharmacist and philosopher) compared results focusing on clarity of the individual findings and degrees of (dis)agreement, dialogically settling on meta-themes. Disagreements were further discussed on a per theme basis, paying explicit attention to preconceived ideas of key concepts such as “ownership” and “autonomy”, triggering reanalysis of the initial coding schemes and themes. As such, triangulation contributed to an additional layer of critical assessment of researcher positionality, resulting in higher-level analysis of the respective themes and promoting the overall richness of the analysis. Finally, feedback on the generated meta-themes was gathered from the respective research groups (VITO and PGL-project group).

## 3. Results

### 3.1. Participants and Demographic Information

An overview of the demographic characteristics of the participants is given in Table 1. Additional information on the IAM Frontier interviewees and PGL focus group participants is found in Table 2. 

#### 3.1.1. IAM Frontier Surveys

Thirty participants were included in the IAM Frontier pilot study and completed both surveys (Figure 3). All participants were healthy and between the age of 45 and 60. The sample consisted of three male and three female participants. Further diversity in gender presentation was not reported by participants. Due to logistical constraints of the pilot study all participants were VITO employees, and as such, highly educated. No relevant differences between (age or gender) groups were found.

#### 3.1.2. IAM Frontier Interviews

From the initial IAM sample, three men and three women were selected, of which three were between 45 and 50 years of age and three were 51 years or older. All six participants took part in both the “pre” and “post” interview. 

#### 3.1.3. PGL Focus Groups 

Six participants were included in the focus group discussion for healthcare users: five women and one man. Five of them were aged 51–70 years, and one of them was aged 30–50 years. Five participants had prior personal experience with genetic tests, either because they had a genetic disease themselves or through their child. The other participant did not have personal experience with genetic tests. Regarding the healthcare professionals’ focus group, four participants—two women and two men—were included. Half of them were aged 30–50, and the other half were 51–70 years. Two participants worked as pharmacists, one as a clinical geneticist, and one as a policy officer of a patient association. 

### 3.2. Meta-Themes and Themes

Four meta-themes were generated following the triangulation protocol (Section 2.4.4): (i) attitudes toward receiving genomic information; (ii) attitudes towards ownership and sharing of genomic data; (iii) autonomy and trust; and (iv) the applicability of the Personal Health Data Spaces. In addition, we identified several subthemes. An overview of the meta-themes and sub-themes in respective datasets is shown in Table 3. 

### 3.3. Attitudes toward Receiving Genomic Information

The first meta-theme concerns attitudes toward receiving genomic information. More specifically, this meta-theme captures how participants related to specific types of sequencing results. Unique to the IAM Frontier setup, participants were questioned prior to and after receiving individual sequencing results, offering a hypothetical and more concrete perspective on the reception of genome sequencing results.

#### 3.3.1. Attitudes toward Hypothetical Genomic Test Results

##### IAM Frontier Surveys

As Table 4 indicates, survey respondents were generally curious about genetic predispositions and stated interest in collecting information on their genome to the greatest extent (*n* = 29). Table 5 shows the survey respondents’ attitudes when presented with various categories of hypothetical WGS findings. Overall, participants were particularly interested in receiving actionable and non-life-threatening (*n* = 29), non-actionable (*n* = 29), and life-threatening non-actionable (*n* = 24) findings. In the case of Variants of Unknown Significance (VUSs), opinions were mixed; half (*n* = 15) of the sample was interested in receiving such results.

##### IAM Frontier Interviews

Interviewees generally related positively to receiving individual WGS results when presented with hypothetical scenarios. Answers differed concerning the specific type of results (i.e., actionable, VUS, non-actionable). All six participants wanted to receive WGS results on *actionable* conditions. The ability to take preventive measures to benefit future health and feeling prepared were offered as reasons. However, many stressed that the type of lifestyle change required to mitigate risks, ranging from exercise to diet and screening to treatment, would be an essential consideration for whether they would want to receive specific results. 

Opinions on being informed about *VUSs* were more varied. Several participants suggested that such findings might cause unnecessary concern due to the uncertainty of the information. Others, however, did express a desire to be informed on VUSs. Several participants noted that while this information might not be of direct benefit, it could still be helpful when developments in genomic medicine or sequencing analysis allow for novel insights, evincing faith in the progress of genomic research.

The interviewees unanimously wanted to be informed of non-actionable findings, as this would allow them to prepare for the future. This wish for preparedness was expressed in different ways. Participants stated that it would allow them to inform and (emotionally) prepare themselves and others. Second, they suggested this would allow them to make conscious decisions to enjoy life more. Furthermore, several interviewees suggested that this information could inform end-of-life decisions. Lastly, some participants expressed optimism that more research might create additional treatment options from which they could benefit after being informed about this genetic predisposition at an early stage. Psychological distress was suggested as a possible negative side effect.

Participants were also asked which specific health-related findings (i.e., diagnoses) they would be most interested in receiving. Overall, participants expressed interest in findings related to familial disease burden and ‘popular’ diagnoses, such as Parkinson’s, Alzheimer’s, and various types of cancer. Furthermore, participants’ interests related to findings were contextualized to their particular social and cultural situations, including age, social status, and familial bonds. 

##### PGL Focus Groups 

Some healthcare users said they would always want to know the results of a genomic test because it is about their health, and they want to be informed and prepared for possible consequences (e.g., the decision to have children). Others said it depends on the actionability of the findings and an individual patient’s situation (e.g., age, potential quality of life implications, and consequences for offspring). All participants agreed that it should always be the patient’s decision what to know and what not to know (i.e., right not to know). 

The clinical geneticist indicated that diagnostic genomic tests only generate results to answer a specific diagnostic question. This method prevents HCPs from knowing information unrelated to the diagnostic question. 

Clinical geneticist (PGL-P8): “You know what? For this patient, I will snoop around the entire genome. That is something we never do. That is forbidden. […] We really want to avoid situations in which we know more than a patient. We have regulations for that.”

Relying on current guidelines and regulations, the clinical geneticist explained that patients sign a “contract” that documents what they wish HCPs to do in case of unexpected incidental findings and of what findings they do or do not want to be informed. This means that current regulations and practice are in accordance with the opinion of healthcare users: the decision of what (not) to know should be the patient’s, and HCPs are obliged to follow the patient’s preferences.

#### 3.3.2. Attitudes after Receiving Individual Genomic Test Results

##### IAM Frontier Surveys

IAM Participants were additionally surveyed on attitudes towards receiving different categories of results after receiving individual sequencing results (Table 6). These results show similar trends. Participants were generally interested in actionable conditions. Results regarding VUSs were mixed. A shift concerning attitudes to nonactionable, nonpreventable results was observed. 

Table 7 indicates that several participants were disappointed in the amount of information received after genome analysis. The interviews provide additional insights here.

##### IAM Frontier Interviews

The interviewees, as well, expressed disappointment concerning the limited amount of (actionable) genomic information that was returned individually after WGS. Reasons included the following: being already aware of potential genetic risk factors due to familial disease burden, beliefs related to the novelty of genome sequencing, and high expectations regarding the comprehensive nature of genomic information. This disappointment is evident in the following quote, in which IAMF-P5 laments that the return of results after sequencing was limited to actionable, health-related findings.

IAMF-P5 (post): *“I mean, there should be more information than just that in my genome*… it might be that there is no defect other than that, but I still want to know if there is any interesting stuff about that, you know.”

### 3.4. Attitudes toward Ownership and Sharing of Genomic Data 

#### 3.4.1. Between Data Ownership and Data Control

##### IAM Frontier Survey 

A second meta-theme was related to data management. When prompted who should be considered the owner of genomic data, most survey respondents (*n* = 23) suggested the donor holds ownership over their (genomic) data. The doctor and the institution performing the analysis were selected as primary “owners” by five and two respondents, respectively. As the interviews indicate, however, ownership is conceived in a myriad of competing ways.

##### IAM Frontier Interviews

The topic of data ownership came up naturally in the interviews. One interviewee referred to the General Data Protection Regulation (GDPR) to argue that all genome analysis results *belonged* to the person that donated the sample. 

IAMF-P5 (post): “We have the GDPR legislation for a few years in Europe now. It basically was a step of defining that anything that concerns the person in terms of information basically belongs to that person, and the *person should have control over how it is shared* through the process and so on. And this should be transparent and all. This is a basic right that Europe has identified for its citizens. I think genome information just falls into that category. It is information about myself and information that I might not even have access to myself due to technical reasons. You know, I don’t have access to all the information that Facebook has about me. Even though I don’t have a Facebook account, they do have a shadow account for me; that’s pretty clear because they spy on everybody. And I have no idea what kind of information they have about me. I don’t think this is the way it should be. Genome or not.”

Several meanings of ownership are intertwined in this quote. First, this participant argues that GDPR grants individual property rights over genomic information. Simultaneously, he stresses the importance of data control and transparency. Further, IAMF-P5 suggests the information is about himself rather than strictly belonging to him. The analogy at the end of the quote provides some clarity. This participant’s comment regarding the social media platform Facebook is used to evoke his concern about the loss of control, transparency, and access. Other participants, too, admitted to complex, nuanced relations with their genomic data. While they indicated a uniquely personal view of genome data, the rhetoric of ownership was rarely utilized. Many, however, stressed the importance of future, on-demand data access. As such, they suggest that the research institute (i.e., VITO) will act as a (responsible) data steward, reliably and accessibly storing their data for them.

IAMF-P6 (pre): “All I hope is that my genome sequence is preserved and if something happens later on and the doctor says: “what’s this here?” Then I can say: “Shall I request it [sequencing data] back, and can you compare it for me?”

While this participant expresses a personal connection to *their* data, they view their sequence as ultimately preserved and managed by the research institute. This dependence, however, does not exclude an ongoing, even personal relationship with their data. What is important for this participant is the possibility of future access to their sequencing data, even if they are not directly in control of their data. Other interviewees expressed similar sentiments.

##### PGL Focus Groups

All healthcare users indicated that patients themselves should be the owners of their medical data. When asked who should be “in charge” of genome data, many, unreflectively, nominate the patient. Again, however, a multitude of meanings regarding ownership was observed. Further scrutiny revealed that healthcare users were particularly interested in data control. One of them added that it would be helpful to have a healthcare professional, a general practitioner (GP), to support interpreting the data. The healthcare professionals agreed with this statement: the patient or its legal representative should be the data owner, but a professional’s support is needed to explain the data and ensure that data management is handled safely and responsibly. 

Healthcare user (PGL-P1): “I think it is about you and everything that comes with it. I think it is good that you yourself *should be in control of it [the data] purely because it is about you*. I think no one should be able to do something with my data without my permission. Simple as that.”

As demonstrated by this quote, the fact that the data is about the patient offers a sufficient argument for data control and involvement in decisions regarding further data use. Based on these and similar statements, data ownership and data control were used seemingly interchangeably by the focus group participants. 

#### 3.4.2. Data Sharing

##### IAM Frontier Survey

Most respondents were willing to share their genome data with family members (Table 8). Scientific researchers were equally deemed appropriate regarding with whom to share genomic information. Sharing with pharmaceutical companies was less well-received. 

As shown in Table 9, attitudes toward data sharing after receiving sequencing results differ from the hypothetical scenarios presented in Table 8, showing more hesitancy to share genome data. After receiving sequencing results, half of the participants reported not having shared genome results with blood relatives; 53% reported having shared results with non-blood relatives. 

##### IAM Frontier Interviews

The interviews provided further insight into these results. Overall, in hypothetical scenarios, participants wanted to share their data with different actors. The degree to which they were willing to share and with whom, however, was intimately connected to the nature of the findings and the actor’s intent. 

In the case of actionable findings, all participants stated that they would inform blood relatives due to the shared nature of these results or, more generally, openness towards those close to them. When interviewees were questioned about sharing findings of VUSs with blood relatives, most participants opted to forego this option to prevent unnecessary worrying. Sharing non-actionable findings with blood relatives garnered varying opinions among the participants. Some would share this information out of a general openness towards family and a moral obligation not to withhold potentially vital information. Unnecessary worrying was again offered as an explanation for reluctance. The interviewees maintained this position after receiving individual sequencing results.

Several participants also responded positively to sharing study results with care practitioners. When asked whether they had brought their sequencing results to a consultation with their GP, several participants mentioned that they did or planned to. Participants generally preferred sharing their information with a healthcare professional who could provide the necessary context for sequencing results (“Yes, I would prefer those things to be more controlled through a GP or hospital or something”, IAMF-P2). This statement also reflects on the topic of data control. Rather than regarding data as being subject to ownership as property, this participant recognizes the value of extending their data relationships to include healthcare professionals. The interviewees also reacted positively to sharing their data in a research context. Several pointed out that the regulated nature of scientific research is vital in these considerations. These views did not change after receiving individual sequencing results.

Overall, interviewees evinced reluctance to share data with unequivocal commercial entities. Several participants suggested that for-profit interests could conflict with the ‘common good’. Others mentioned the lack of regulation of private companies and the risk of data breaches as barriers to sharing genomic information. One participant suggested that genomic data could leak to insurance companies, leading to a (further) “desolidarization” of healthcare—evincing worries of broader societal impact of an unregulated, market-driven implementation of genomic medicine. Furthermore, participants repeatedly stressed the need for contextualization of sequencing results and suggested that commercial entities would not be able to offer this. 

IAMF-P2 (pre): “I have a bit of a problem with that [sharing with commercial entities] because their angle is more commercial, and if there is insufficient control over that. Certain decisions may not be taken in the interest of the group of people but rather in the interest of the commercial perspective of that organization. Then, in this sense, I think the risk is a bit too great to leave that to them.”

Again, concerning data sharing with commercial entities, this participant stresses the importance of (institutional) data control, stating that commercial entities operate on for-profit incentives and currently lack sufficient regulatory oversight to be trusted with personal genomic data. Regulatory frameworks are deemed necessary to curtail a (purely) market-driven implementation of genomic medicine. According to participants, the risks of regulatory neglect might hold (negative) consequences for society as a whole.

##### PGL Focus Groups

In the healthcare user focus group, all participants said they would discuss their genomic findings with their children. They would, however, first ensure whether their children would want to be informed about these results and whether they would want to be tested themselves. 

Most healthcare users said to trust their HCP with their data. However, two participants indicated that they trusted no one with their data. They feared sharing their medical records would cloud HCPs’ judgment leading to false assumptions and prejudice. Both said that prior negative experiences with healthcare professionals had harmed their trust. Most other healthcare users recognized this. Many healthcare users said they were pushed from pillar to post, not taken seriously (e.g., being labeled as an overly concerned mother), and did not have access to all data (e.g., feeling data was withheld from them) in previous encounters with HCPs. This led some of them to keep an administration of their own or their child’s medical files themselves.

Additionally, some said that they wanted something in return for sharing their data (PGL-P4: “We are not just research material”). This includes being informed on the latest medical developments that might be relevant to their condition. Instead of a one-way interaction, they want to be notified, taken seriously, and work together. This participant indicates a need for participation and communicates a worry for being “used” for research purposes. The healthcare professionals agreed that trust is vital to sharing (genome) data. The healthcare professionals trusted other certified HCPs with patients’ medical data. One of them said that this does not mean that every HCP should have access to all of a patient’s medical data: HCPs should have access only to the specific data needed in the given clinical context. All healthcare professionals were against sharing medical data with commercial and insurance companies. Concerning insurance companies, however, they said it is justifiable for insurance companies to have at least some data, but they were unsure to what degree. 

##### Benefits and Risks of Sharing Data 

Regarding the benefits of sharing medical data, both healthcare users and healthcare professionals indicated that sharing data would increase the availability of knowledge, complete medical records, better and more efficient healthcare, and possibilities for tailoring treatment. 

Clinical geneticist (PGL-P8): “Benefits of data sharing include more efficient and better healthcare. No unnecessary tests, no need for long waits on results that are collected elsewhere.” 

Healthcare users named continuity of care (not having to repeat their story, risk of forgetting something) as an additional benefit. Regarding risks, all participants named the risk of data misuse, privacy concerns, and data leaks. Additionally, healthcare users named high costs and the risk of false assumptions due to prejudice or incomplete records.

### 3.5. Autonomy and Trust

#### 3.5.1. Autonomy as a Central Value

Autonomy was the third central meta-theme in the interviews and focus groups. In particular, autonomy was primarily conceived in terms of (individual) choice. Participants deemed autonomous choice as trumping societal benefits of genomic medicine—it should always be the individuals’ choice to have their genome sequenced. Concerning sharing genome data, they suggested that the donor should be the center of decision-making. We further inquired about autonomy within genomic medicine concerning those (i) undergoing sequencing and (ii) receiving sequencing results. 

##### IAM Frontier Interviews

Interviewees stressed that everyone should be free to decide whether they want to have their genome sequenced. Generally, perceived societal (health) benefits of implementing genomic medicine on a large scale were acknowledged, but everyone stated that this could not transcend individual autonomy. Several participants, however, suggested that policy measures, such as reimbursement through health insurance or cost decreases, could stimulate or nudge them toward the pursuit of genome analysis. 

Concerning receiving specific results, participants further stressed the importance of individual autonomous choice. In all situations—actionable, non-actionable, and VUS results—and despite generally positive attitudes towards receiving results among participants, the interviewees suggested that the patient should always be central in decision-making about which WGS results they wanted to acquire. Versions of a right-not-to-know were leveraged in relation to sharing sequencing results with blood relatives. 

##### PGL Focus Groups

The healthcare users all agreed that it should be the patient’s decision what (not) to know. Additionally, they all wanted to be thoroughly informed, have access to their data, and be involved in decision-making (“It all comes down to the fact that you decide for yourself and not that somebody else decides for you”, PGL-P2). One healthcare user suggested capacity to be an important consideration for autonomous decision-making: 

PGL-P1: “This doesn’t hold if you’re not capable of making decisions about your own data, but otherwise I assume that every human being is able to make normal decisions about their data and what to do with them, for their own good. So yes, I would say that is up to the person who is capable of saying something about them [data] and doing with them.”

As such, in emphasizing autonomy, this healthcare user suggested that “capacity” might pose limits to individual decision-making. It is unclear, however, which capacities this participant has in mind. 

#### 3.5.2. Limits to Autonomy and the Importance of Trust in Genomic Medicine

Despite the importance of autonomy, as demonstrated in interviews and focus groups, participants often situated their autonomy in a broader social and cultural context. Conversely, participants acknowledged that individual autonomy in genomic medicine could be bounded and limited by social and relational considerations. 

For one, many interviewees admitted feeling obligations toward family members to disclose results. Genetic kinship and, as such, direct but also indirect (psychosocial) impact was found to produce such moral imperatives. This worry is expressed vividly in the following quote: 

IAMF-P5 (pre): “Yes, of course, I would want to know. […]. I would basically *need to inform everybody who is affected*. […] I would think that it is basically kind of *an obligation to spread that information* […]”

Second, participants suggested that the sensitivity of medical data involves particular vulnerabilities. Therefore, many expressed the need for data safety structures and regulations as preconditions to engage in data relations of stewardship. Third, the complexity and context-sensitivity of sequencing results caused most participants to suggest the necessity of involving healthcare professions. Finally, and related, low genetic literacy in the general population was presented as a potential justification for (certain) restrictions to widespread, unregulated genome sequencing, i.e., limiting individual autonomy, for example, through restricting access to direct-to-consumer (DTC) genome testing. 

Given these limitations to individual autonomy, participants across cohorts recognized trust as an essential mediator in genomic medicine. Several levels of trust were identified across surveys, interviews, and focus groups, including (i) institutional trust, (ii) trust in science, and (iii) interpersonal trust in healthcare professionals. 

##### IAM Frontier Survey

Table 10 shows survey responses concerning the implementation of genomic sequencing at different societal levels. Opinions among respondents were somewhat mixed. In general, e-commerce and other commercial settings (including pharmacies) were deemed less appropriate to provide access to genome analysis. Survey respondents seemed to prefer institutional health services, such as the hospital setting and the requirement of a doctor’s prescription as appropriate in implementing genomic medicine. The interviews provide additional information here. 

##### IAM Frontier Interviews and PGL Focus Groups

As interviewees indicated, institutional trust was a significant factor in participants’ deliberation of genomic medicine. A lack of trust was repeatedly identified as a potential barrier to genomic medicine across various societal levels. For one, interviewees suggested that regulatory standards and robust, safe data infrastructure drove their preference for limiting the implementation of genomic medicine to existing institutional settings. Conversely, commercial entities were deemed inappropriate and untrustworthy due to (potential) data breaches and for-profit motives. The lack of existing legislation was offered as a reason not to trust the commercial sector with sequencing data. Other factors impacting trust included transparency and adequate care support systems, such as opportunities for additional genetic counseling.

Interviewees’ attitudes towards the uncertainties encompassed in the technological and interpretative limitations of genome sequencing were often supported by trusting attitudes concerning science. Many quoted the supposedly inevitable progress in the interpretative capabilities of genome sequencing results. 

IAMF-P2 (pre): “I see it this way: It is useful that I get this information, although I can do little with it myself. But if it turns out that the studies continue or that people will know more in 5 or 10 years and then it comes to light that I have that gene, it is useful to have that information about myself.”

Despite the disappointment in the limited availability of meaningful information after genome sequencing discussed earlier, many—as did this participant—suggested that as science progresses, current uncertainties (i.e., VUSs) might, nevertheless, provide additional information on their individual risk profile.

Interpersonal trust in healthcare professionals was also identified as crucial for genomic medicine. In the focus group study, healthcare users and professionals indicated that trust is vital for the willingness to share information and thus functions as a necessary condition for genomic medicine. However, as mentioned in “PGL Focus Groups” section, negative prior experiences with HCPs can significantly harm patients’ trust in (future) HCPs. 

IAM Frontier participants reported high degrees of trust in HCPs and they preferred HCPs to be the primary point of contact in genomic medicine due to the sensitive and complex nature of genomic information and the possibility to provide additional support and psychological counseling. Interpersonal trust in healthcare professionals was reported to be related to physician competencies, including adequate knowledge about genomic medicine and the communication skills necessary to relay the nuances of genomic findings. Interestingly, participant opinions swayed in the second set of interviews, after receiving individual sequencing results from the study doctor. Many remarked that the study doctor—a primary care physician—lacked some of the necessary competencies and sensibilities to report sensitive genome data. Participants, as such, suggested that the importance of genetic literacy is not merely crucial for citizens but equally extends to healthcare professionals.

IAMF-P1 (post): “So I do think it is very important to put results in the right context. And, as I say, not everyone is able to interpret those results correctly. So I think it’s very important that they are framed correctly by someone *who is qualified to do so*. That is probably just as important as the results themselves.”

All participants favored additional support services beyond the current primary care landscape, such as genetic counseling, to contextualize patient genome information. They noted, however, that engagement with these services might depend on individual preferences and prior knowledge related to genome sequencing.

#### 3.5.3. Tension between Patient Protection and Patient Autonomy

##### PGL Focus Groups

An interesting tension was observed between healthcare professionals’ duty to care and patients’ individual autonomy. While the healthcare professionals stated the importance of patient support in understanding and handling genetic data and protecting patients from potential harm, the healthcare users stressed the importance of patient autonomy. 

To understand genetic data, raw data need to be translated into interpretations and explained in the specific context of the patient. Due to the complexity of these data, all healthcare professionals stated the importance of supporting patients in understanding their genetic test results and their consequences. Furthermore, due to scientific developments and the sensitivity of genomic tests, the interpretation of raw data can change over time. Therefore, genetic counseling and follow-up were considered key. Additionally, healthcare professionals pointed to the risks of misinterpretation among patients. One healthcare professional, for example, was worried people might stop seeking medical support when their genomic results ‘look good’ or, vice versa, worry unnecessarily. This duty to prevent potential harm as a result of patients’ limited understanding of genomics and the specific context in which the results should be interpreted is illustrated by this quote:

Clinical geneticist (PGL-P8): “*We must protect people from that*. […] For example, when someone says: ‘There is nothing wrong with my DNA, so I don’t have to go to the doctor’. […] So, you want people to understand the context in which the results are interpreted.”

The risks of patients’ limited understanding were not a central topic in the healthcare users focus group. The importance of patient support in understanding and handling genetic data was recognized by some healthcare users. However, as demonstrated earlier, patients sometimes have trouble trusting HCPs due to prior negative experiences. Additionally, all healthcare users stated that patients should be in *control* over their data. 

### 3.6. The Applicability of Personal Health Data Spaces 

The final meta-theme related to the applicability of PHDSs included the sub-themes (i) characteristics of a PHDS, (ii) functionalities of a PHDS, (iii) data types to be stored in a PHDS, (iv) implementation of PHDSs, and (v) the potential of a PHDS. 

#### 3.6.1. Characteristics

Healthcare users and healthcare professionals were able to come up with a definition of a PHDS that was close to the next-generation examples described in the introduction. They described a PHDS using the following characteristics: (i) a secure environment, (ii) owned and managed by the patient, (iii) makes medical data available and accessible to patients, (iv) saves and stores data in one place, and (v) allows patients to consent to their data being shared with HCPs. 

#### 3.6.2. Functionalities 

In addition to these characteristics, healthcare professionals liked the functionality of adding analytic tools, such as data interpretation support, to help patients understand their results. They also said that they would like a PHDS to support HCPs in signaling and communicating genetic vulnerabilities. For example, when new developments become available in genomics tests and the interpretation of the test results, patients could receive a notification advising them to make an appointment with their clinical geneticist or other HCPs. This would not only be efficient for HCPs, but it would also support patients in knowing if and when it would be helpful to schedule a follow-up appointment. Healthcare users also stressed the importance of being informed of relevant medical or scientific developments:

Healthcare user (PGL-P4): “It should be standard practice that when a new mutation is found, the ones who have made their data available should be informed.”

This quote illustrates the desired ‘two-way’ interaction when it comes to data sharing: patients share their data and in return they want to be informed about relevant results. By using the notification function mentioned above, PHDSs could facilitate this process of informing and involving patients. 

Additionally, the healthcare professionals would like a PHDS to be able to share patients’ family histories. Currently, as illustrated by this quote, patients are dependent on their family members to inform them about relevant genetic vulnerabilities:

Policy officer patient association (PGL-P9): “I think that would solve a current problem. People are dependent on others whether information reaches them.”

This problem of inaccessible health information, due to dependence on other people, prevents people from being in control over their data and limits their autonomy. Sharing and saving patients’ family history in a PHDS would prevent data from getting lost and could make patients less dependent on their family member. In doing so, PHDSs would increase data control and autonomy. 

Another functionality mentioned by a few healthcare professionals was the possibility of building “walls” in a PHDS to prevent unwanted actions, such as sending data to unreliable parties. In this regard, this functionality could support healthcare professionals in their duty to prevent potential harm. Additionally, healthcare professionals thought that these “walls” could prevent access to genetic results that the patients have indicated they do not want to see. Thereby, PHDSs support patients being in control regarding the specific data of which they do or do not want to be informed. 

#### 3.6.3. Type of Data That Should Be Stored in a PHDS

An important theme involved the specific data types that should be stored in a PHDS. All healthcare professionals and some healthcare users said that the data in a PHDS need to be living data responsive to medical developments. What this would look like, however, is highly complex and gives rise to a range of questions, as evinced by the following quote: 

Policy officer patient association (PGL-P9): “The question is: What data would you place in the locker? Which information would and wouldn’t you place there? The raw data, so that the patients can take these data for further analyses? Or only the results? And what to do with incidental findings patients have indicated not wanting to see?”

As illustrated by this quote, the healthcare professionals thought it was important to think of the specific type of data that should ideally be stored in the PGL but struggled to answer this question. Would this be the raw data, only the results, or the whole genome? Additionally, this question raised many practical and ethical follow-up questions, for example, regarding incidental findings. All healthcare users stated the need for agreements to prevent patients from unwantedly accessing incidental findings, thereby highlighting healthcare users’ need for control and their ability to make autonomous decisions regarding the data of which they do or do not want to be informed. 

Some healthcare professionals considered storing complete genomic sequences in a PHDS:

Pharmacist (PGL-P10): “I had the idea that it would be this way in the future, that your whole genome would be in the data locker. […] And when you see a doctor for a certain problem, without needing additional tests, you could see what your DNA contributed. In principle, you could test once. And then you save it all in the locker, and the next step is that it should be interpretable at a certain moment.”

This quote illustrates that by saving WGS data in a PHDS, a PHDS would facilitate the use of genomic information to answer diagnostic questions, thereby promoting the continuity of care. 

#### 3.6.4. Implementing a PHDS: Barriers and Facilitators

The healthcare professionals listed several barriers and facilitators for implementing a PHDS in healthcare (Table 11). Additionally, some healthcare users pointed out that in the presented PGL scenario, we assume that every individual has genomic data. Still, most people do not have this kind of data. Subsequently, they wondered how individuals could get access to genomic tests. 

Additionally, as stated by a healthcare professional, the general public needs to accept the concept of a PHDS.

Pharmacist (PGL-P10): “PGLs need to be accepted. People should have the idea: this is nice to have or a need to have. That realization needs to sink in.” 

According to this participant, the general public needs to understand how patients could benefit from a PHDS in order to accept tools, such as the PGL. Another important facilitator that was named is education. HCPs and patients need to be educated on PHDS tools and sufficiently trained on how to use them.

Policy officer patient association (PGL-P9): “The rollout is quite a task, I think, also regarding communication and explanation and ensuring that everyone knows how to work with it.” 

Hence, acceptance and education are essential to facilitate the implementation of PHDSs in healthcare.

#### 3.6.5. Potential of a PHDS in Healthcare 

All healthcare users were in favor of having their own PGL. The most reported reason was that participants believed a PHDS would put them in control of their data without any interference or dependence on HCPs, hospitals, or institutions. 

Healthcare user (PGL-P4): “I would like to manage my data myself.”

This management concerns available and accessible data to the patient themself and being able to decide what types of data are stored in the PGL and what types of data are shared with HCPs. 

A final consideration that a few healthcare users mentioned was that the PGL would prevent data from getting lost. 

Healthcare user (PGL-P4): “I was tested in 2000. At the time, there were only paper files in that center. […] But now, no one can find my DNA data anymore. For me, that alone would be enough to want my own locker.”

This quote not only highlights the need for control but also concerns about continuity of care. After all, lost data mean a need for repeated tests, which disrupts and delays patients’ treatments. In line with the suggested functionalities of data interpretation support and notifications, other reasons were that the PGL could provide additional insight into the data, which could support them in staying up-to-date with relevant medical developments, in line with Section 3.6.2. with respect to suggested functionalities of data interpretation support and notifications. 

All healthcare professionals were in favor of implementing a PHDS. The main reason was that it would facilitate the use of genomic information in clinical practice; it would improve data availability and make the process more efficient. 

Clinical geneticist (PGL-P8): “I know for my little piece of patient care, if the PGL would already exist, it would be useful. For example, a genetic test was done in Leiden. The patient moves, and four years later, I see her in my office. I go through all the paperwork. And before I have the results, it is days or weeks later, without anyone actively trying to delay things. And now, I click and there it is on my screen. I can immediately help my patient.”

Pharmacist (PGL-P9): “I agree. At this moment, patients come with a card on which it [results] is written and, of course, we could enter this in our system. But in fact, it is only the translation. What does it mean? […] When you are more of an expert as a pharmacist, you would like to have the raw data or see the tested variants.”

These quotes illustrate that the current process of accessing and using genomic data is considered slow and inefficient, imposing barriers to adequate care. The PGL was thought to speed up the process and provide rich information to HCPs. 

## 4. Discussion

Advocates position genomic medicine to offer important benefits for individual patients, healthcare professionals, and the healthcare system as a whole. However, several challenges—ethical, conceptual, technological, and practical—need to be addressed to ensure proper implementation. This study aimed to explore the perspectives of a diverse range of stakeholders on genomic medicine and personal health data spaces. Using a mixed-method design, healthy participants, healthcare users, and professionals were asked about attitudes toward genomic findings and genome data. 

### 4.1. Attitudes towards Receiving Genomic Information 

The current literature indicates the need to gain additional insight into citizens’ attitudes towards genomic findings and personalized medicine [5,10,11]. Our first meta-theme investigated participants’ attitudes towards various types of genomic findings.

Across datasets, participants expressed high interest in receiving individual genomic findings. While most participants were particularly interested in receiving actionable findings, hesitancy was expressed in the case of VUSs. These findings correspond to those of Middleton et al. [49]. Across cohorts, participants expressed a strong interest towards receiving non-actionable individual sequencing results, echoing other findings in the literature [50]. 

Participants offered several reasons to support their interest in receiving individual findings. Concerning actionable results, IAM Frontier participants quoted controllability and predictability as the primary concerns. Treatability, controllability, and predictability as significant motivators for interest in sequencing findings were reported in earlier studies [50,51,52,53,54]. Interestingly, several participants considered the impact of genomic findings in relation to end-of-life decision-making—signaling the normativity and situatedness of actionability beyond the definitions operationalized in the ACMG recommendations for reporting individual findings. Furthermore, participants across cohorts stressed the potentially negative psychological impact of receiving non-actionable and VUS results matching similar findings in the literature [53,54]. Positive attitudes towards VUSs found in the sample were often related to a sincere faith in scientific progress. Participant beliefs in this supposedly straightforward increase in genomic interpretative capabilities echo reports in the literature [53,54,55]. However, it is unclear to what extent such convictions are substantiated [56], indicating the need for improved genetic and scientific literacy [50,57,58]. 

The necessity of transparency and education based on the potential and limitations of genomic medicine is vividly apparent in the sense of disappointment in the IAM Frontier cohort after receiving individual sequencing results. The prevalence of high expectations about genome analysis in our cohort is consistent with other studies [59,60,61,62]. While negative attitudes in genomic screening programs with healthy participants have been described, most studies report primarily positive reactions to receiving information [63,64,65]. However, most reports in the literature solely inquire about hypothetical scenarios, offering a possible explanation for these contrasting outcomes [66]. Moreover, since the IAM Frontier sample was exceptionally well-educated and research-minded, these findings further stress the urgency of improving patient education and genetic and scientific literacy. Furthermore, participants’ disappointment echoes suggestions in the literature that access to health-related information does not necessarily amount to a sense of empowerment. Our findings suggest that individual attitudes towards individual findings are related to their novelty, usability, and meaningfulness [67], providing further support for patient-driven conceptions of empowerment [19].

### 4.2. Complexities of Genomic Data Management beyond Ownership 

A second meta-theme persistent across datasets handled participants’ relations with genomic data management. 

When asked to designate the *owner* of genomic data, most participants elected patients as the likely owner of their medical data. These results echo other citizen engagement projects in the Belgian population [68,69]. 

However, recent legal and ethical scholarship on genomic data management suggests that expressions of ownership span multitudes of meanings and “competing narratives” [70] beyond mere property claims [70,71,72]. Despite this conceptual murkiness, appeals to ownership often function to express a range of genuine concerns worthy of appreciation. We argue that refraining from taking “ownership” claims at face value and focusing on concerns participants might have about their data provides more comprehensive and nuanced insights into perspectives on genomic data management. We offer three concerns evinced within our cohorts relevant to this debate: data control, responsible data stewardship, and responsibilities towards others.

First, whereas participants often did employ ownership rhetoric, their concerns about genomic data management are better captured in terms of data control. This murkiness between ownership and control was particularly present in one interviewee’s appeal to GDPR as safeguarding data ownership and alleviating the lack of data control and access. While GDPR aims to ensure data protection, it does not provide property rights over data [72]. Moreover, currently, it is unclear to what extent GDPR applies to genome sequencing data protection [73]. Further scrutiny reveals that this appeal to the legislative context serves as a means to reinstate a sense of control over genomic data. Several other participants provided similarly vivid expressions of concerns related to genome data management. Strikingly, in appeals to “ownership”, multiple participants suggested genome data are “about me”, rather than “belong to me”. As ethicist Angela Ballantyne argues, expressions of “aboutness” signify the special relationship people have with personal data while simultaneously capturing concerns about being disconnected from it [71]. As such, while participants in our cohorts generally feel genomic data have a uniquely personal quality, they are apprehensive about losing data control.

Second, participants recognized that data management (including data sharing) necessarily involves a range of actors and corresponding responsibilities. For one, interviewees stressed institutions’ responsibilities to provide transparent, safe, and free access to genomic data. It is unclear, however, which conception and what level of transparency is operationalized by our participants. Minimally, they refer to some level of insight in their data either on an individual level or in conjunction with an HCP. Further research on the desirability of different types of transparency is warranted. While participants expressed willingness to share data with research, this inclination was premised on several conditions. For one, healthcare users indicated that data sharing should be a two-way interaction. Participants expect to be thoroughly informed of the results, taken seriously, and treated as partners in decision-making. A bidirectional flow of information between participants and research institutes is increasingly considered standard practice in participatory research paradigms, including biobanking, precision, and genomic medicine [74,75,76]. Importantly, IAM interviewees professional familiarity with scientific research might lead to overestimations on the benevolent intent of scientific research(ers). 

Hesitancy to share with commercial entities was expressed across cohorts due to worries about data breaches, for-profit motives, lack of regulation, and a negative societal impact on healthcare. These findings correspond with other citizen engagement projects on genome data in Belgium [68] and stress the importance of trustworthy, transparent data infrastructures. 

Despite these risks, healthcare users and professionals acknowledge the potential benefits of robust data-sharing structures for healthcare efficiency and quality. It could increase knowledge availability, lead to more complete medical files, and support personalized treatment. A review of health information exchange supports these benefits: sharing information leads to improved patient safety and increased efficiency, as demonstrated by fewer duplicated procedures, reduced imaging, and lower costs [77]. However, such benefits of data sharing for healthcare practice are premised on (institutional and commercial) actors acting as responsible data stewards [70]. 

A third concern expressed by participants pertains to genomic data and individual responsibilities towards others. Most participants indicated willingness to share genomic information with family, which was in accordance with previous studies on the Belgian population [78]. The relational nature of sequencing results—i.e., these data are also about someone else—was often cited as producing a range of moral imperatives to (refrain from) sharing results with genetic and nongenetic kin [69]. Participants found that genomic data management involves a range of individual (and familial) responsibilities—introducing some interesting tensions with their (individual) autonomy-oriented attitudes towards genomic medicine in general, discussed in the next section. 

### 4.3. Autonomy, Choice, and Its Limits

Participants often stressed the individuals’ choice in genome sequencing and data control. This focus on autonomous choice within cohorts of citizens in genome sequencing was established by earlier research in the Belgian population [68,69,79]. Participants emphasized the significance of sufficient and meaningful information, as well as the capacity to make informed decisions, in the context of genome sequencing, aligning with conventional notions of autonomy and informed consent in bioethics. The value of autonomy, then, was primarily emphasized by participants in terms of autonomous choice raising questions on the more comprehensive conceptions of autonomy embedded in claims of partnership, empowerment, and participation [18].

Despite this emphasis on autonomy, participants often admitted to limits to individual autonomous choice, including (i) the asymmetry in (clinical) care relations, (ii) negotiating between the sensitivity and usefulness of medical data, (iii) the lack of genetic literacy in the general population, and (iv) (moral) obligations towards others as evinced in our discussion on data sharing [80]. These findings echo reports on the need for more comprehensive, relational conceptions of autonomy [81] in genomic medicine [82,83,84]. Our findings, similarly, suggest that participant decisions and attitudes about genomic medicine and data are constructed within and unfold through a matrix of familial, social, and care relations. Rather than retaining an individualistic framework of autonomy, genomic medicine involves the shared, socially-embedded nature of decision-making and, crucially, the importance of trusting relationships and trustworthy institutions serving as enabling conditions for autonomous decision making [85]. 

### 4.4. Levels of Trust in Genomic Medicine

Given the limitations to individual autonomy manifesting in our cohorts, *trust* was deemed to be of central importance to all participants. Following Myskja and Steinsbekk [86], we take trust here to be primarily rooted in vulnerability and built on the assumption of the trustee taking responsibility for the individual’s interest. This conception of trust corresponds to the worries expressed by participants. Moreover, the “default attitude” of trust found in many (though not all) participants in our cohort suggests a pragmatic attitude necessary in the complexities in genomic medicine [86,87]. 

Importantly, we distinguished three discrete levels of trust relevant to participants in genomic medicine. First, institutional trust was leveraged as a precondition for the proper implementation of genomic medicine. Participants favored the implementation of genomic medicine in existing clinical and healthcare settings over commercial settings, including regulatory safeguards, such as the need for doctor’s prescriptions, which correspond with earlier surveys in Belgium and the Netherlands [50,51]. Our findings suggest institutional trust to be an essential consideration for implementing genomic medicine. 

Second, survey and interview participants leveraged trust in science to manage the (clinically) uncertain nature of genomic information. Many participants implied that trust in future developments in genomic medicine motivated them to participate in genome sequencing, even if current findings are limited [88,89]. Again, attitudes towards science in IAM participants in particular might be shaped by their familiarity with the research context. Although these expectations could lead to harm and disappointment when unrealistic, trust in science seems to be an essential factor for the uptake of genomic medicine. As was recently suggested, participatory approaches, such as citizen involvement both in trial design and the development of ELSI frameworks, can foster and maintain trusting relationships in the scientific endeavor necessary for proper implementation [68,90]. 

Third, interpersonal trust in healthcare professionals was considered crucial for genomic medicine. In general, participants trusted healthcare professionals to guide and support them in genomic medicine. Moreover, they stressed the importance of the right competencies, including adequate knowledge, communication skills, and additional clinical support, such as genetic counseling. This is particularly important in light of recent reports questioning the adequacy of practitioner knowledge of genomic medicine [7,91,92,93]. Our findings then indicate participants as well, suggesting a need for genetic literacy and additional education for healthcare professionals. Furthermore, while trust levels in healthcare professionals are generally high in the Low Countries [73], our findings stress the importance of maintaining trusting patient–physician relationships as a continual challenge for genomic medicine. Moreover, it needs to be emphasized that trust in HCPs and (biomedical) research is not uniformly distributed and may differ significantly based on prior (negative) experiences or historical injustices [66].

Our findings suggest that in a healthcare climate dominated by technological (and biological) complexities, a trusting, transparent and enabling infrastructure (through regulatory, competency, and accountability structures) is necessary to garner a form of reflexive, active trust as a precondition for decision-making [86]. This approach acknowledges the complexities of modern healthcare and the importance of shared decision-making, while avoiding a return to paternalistic healthcare practices.

### 4.5. Tension between Patient Protection and Patient Autonomy

Despite these demands for trusting infrastructure, patient autonomy might still be under pressure due to (latent) paternalistic attitudes in individual HCPs. One of the most striking differences between the healthcare users and the healthcare professionals was the tension between patient protection and patient autonomy. 

The healthcare professionals felt a strong urge to protect and support patients, especially patients with low health or digital literacy. They thought explaining, framing, and contextualizing (genomic) test results and their consequences were of utmost importance. This felt responsibility to protect patients and prevent future medical harm as a duty of care (i.e., the bioethical principle of beneficence) was also found in a Belgian focus group study on clinical geneticists and genetic counselors [82]. These professionals felt that this duty to prevent harm outweighed patients’ autonomy in certain situations. After all, patients who do not fully understand the meaning of genomic results and their consequences are limited in their ability to make informed and autonomous decisions [82]. Furthermore, healthcare professionals in the current study felt a responsibility towards their patients to ensure data management is handled safely and responsibly (e.g., preventing data from being shared with untrusted parties), which is supported by studies indicating a need for further public education on health information management [94].

On the other hand, the healthcare users indicated that they wanted to be less dependent on HCPs. Instead, they wanted to be in control. The most prominent reason was that they did not always trust HCPs due to prior negative encounters with HCPs. Consequently, when patients’ trust is damaged in previous encounters, this affects how patients will approach future HCPs. In other words, HCPs must earn their trust [95,96].

As is evident from the above, there is a genuine tension between this “soft paternalism” [82] of HCPs and the generalized need for autonomy expressed by healthcare users. Importantly, this tension presents a significant challenge to one of the central aims of personalized medicine, patient empowerment. Further research must explore the interface between the duty of care and patient autonomy. Our findings support more nuanced, complex, and relational understandings of autonomy explored in the previous section. Another fruitful avenue to overcome this tension is offered by Sandman and colleagues [97] who opt for a dynamic process of shared decision-making in which both the patient and the professional participate actively. When collaboratively reaching a decision, the value of autonomy and beneficence can be respected [97,98]. A PHDS, such as the PGL, could possibly facilitate the process of (shared) decision-making by providing comprehensive and relevant information to the relevant parties. 

### 4.6. Implementation: The Applicability of Personal Health Data Spaces

Part of this collaborative effort was to explore such a Personal Health Data Space solution for genomic medicine. The Personal Genetic Locker project aims to address many of the challenges identified above, including providing safe, robust data control and data management for patients and potentially increasing autonomy. As such, participant attitudes regarding the PGL are an application of the ideas explored above in a concrete PHDS case. 

All participants were in favor of implementing a PHDS in clinical care. Healthcare users especially liked to be in control over their data without any interference from or dependence on HCPs, hospitals, or institutions. In this manner, the PGL corresponds to healthcare users’ concerns about control and autonomy. Additionally, by giving informed consent, healthcare users determine which parties they trust in terms of data sharing. The healthcare professionals thought that the PGL could improve the availability and use of genomic data in clinical practice by making the data-sharing process more efficient. 

To improve the implementation of a PHDS, several facilitators were listed by healthcare professionals: (i) cooperate with similar projects; (ii) involve end-users in the development of the PGL, which was found to result in a better user experience and adoption of new tools [99,100]; (iii) have an elaborate testing phase; (iv) link the PGL to existing EHRs, which is thought to facilitate the registration and consultation of genomic data and offers the possibility of sending automatic alerts when certain combinations of symptoms and familial or genomic risk factors indicate a referral to a clinical geneticist or other medical specialists [8]; (v) educate HCPs on the benefits of PHDSs and how to use them in clinical practice, which is crucial for the adoption of genomic medicine and more appropriate and timely referrals [7]; and (vi) educate the general public on the benefits of using genomic data and the use of PHDSs, which is thought to increase public understanding and address potential concerns, for example, about data sharing [14]. 

### 4.7. Limitations and Strengths

Our findings should be interpreted in light of several limitations. The first limitation concerns the fact that in the study designs, data control and data ownership were not sufficiently disambiguated, potentially increasing the murkiness between these terms in the results. The second limitation concerns the triangulation process. While triangulation was utilized to improve analysis in terms of depth and richness, it potentially introduced some limitations. For one, triangulation was limited by slight differences in the research design and scope, limiting the ways with which the different datasets were engaged. Moreover, with regard to some aspects of the analysis, some datasets were privileged during triangulation potentially effacing nuances found in other data. Finally, differences in methodology (reflexive TA and coding-reliability TA) might limit the fit between datasets.

The third limitation concerns the samples of the studies. Although maximum diversity was strived for, contextual factors constrained this aim. For one, participants in the survey and interviews were highly educated and working in a research environment, introducing potential positive biases. Moreover, in the focus group study, a more diverse group of participants would have garnered a richer analysis. Despite our efforts to include participants both with and without prior experience in genetic testing, most included healthcare-user focus-group participants who had experience with genetic testing, which could have led to limited engagement with the experiences of views of people without genetic testing. This is possibly a result of selection bias, with people with personal experience in genetic testing or genetic diseases being more motivated to participate in such research. In the focus group with healthcare professionals, we failed to include non-specialists in genetic care, such as primary care physicians or other generic healthcare specialists. Although we contacted many non-specialists in genetic care, they declined participation due to a lack of time.

Although the individual studies might have lacked sufficient diversity among participants, by combing the three studies, this paper was able to provide a broad perspective of a diverse group of stakeholders, including healthy individuals, healthcare users, and healthcare professionals—one of the strengths of this paper. Especially regarding the applicability of a PHDS, it was necessary to include both the perspective of the patient and that of the professional since they will use the PGL in different settings. Another strength consists of the IAM Frontier study design, in which the same participants were interviewed and surveyed both before and after receiving their genomic test results. This provided additional insight into attitudes towards genomic medicine compared to the largely hypothetical inquiries in the literature. Finally, by engaging with diverse participants, including healthy citizens, healthcare users with prior experience in genomic medicine, and healthcare professionals, we garnered valuable perspectives on genomic medicine and features for PHDSs. 

## 5. Conclusions

This study explored the attitudes and concerns of healthy participants, healthcare users, and professionals regarding genomic medicine and PHDS initiatives, such as the PGL using a mixed-method approach that included a questionnaire survey, interviews, and focus groups. Triangulation generated four meta-themes: (i) attitudes toward receiving genomic information; (ii) attitudes towards ownership and sharing of genomic data; (iii) autonomy and trust; and (iv) the applicability of Personal Health Data Spaces. Most participants expressed high levels of interest in learning about their genome. Additionally, participants often voiced a sense of ownership. However, closer scrutiny reveals that they were mainly concerned with data control. Most participants acknowledged the benefits of data sharing (e.g., better and more efficient healthcare). Nonetheless, they worried about data misuse, privacy, and the possibility of false assumptions in subsequent clinical visits. For patients to be willing to share information, trust was found to be vital and thus functions as a necessary condition for genomic medicine. More specifically, we identified three levels of trust with regard to genomic medicine: interpersonal, institutional, and scientific. In addition to trust, autonomy was found to be an important meta-theme. All participants thought autonomous choice to be of utmost importance, yet limited by several social and relational considerations. Moreover, healthcare professionals felt responsible for protecting and supporting patients to prevent possible harm, possibly limiting patients’ autonomy. PHDS initiatives were thought to support the adoption of genomic medicine by facilitating the access to and use of genomic data. Moreover, since the patient manages a PHDS, it was thought to enhance patient’s autonomous decision-making. Both healthcare users and professionals were enthusiastic about implementing PHDSs in clinical care and listed several potential barriers and facilitators.

To conclude, our findings provide helpful support and insights for the implementation of and further research on genomic medicine and PHDS solutions in current healthcare systems. 

## Figures and Tables

**Figure 1 genes-14-00786-f001:**
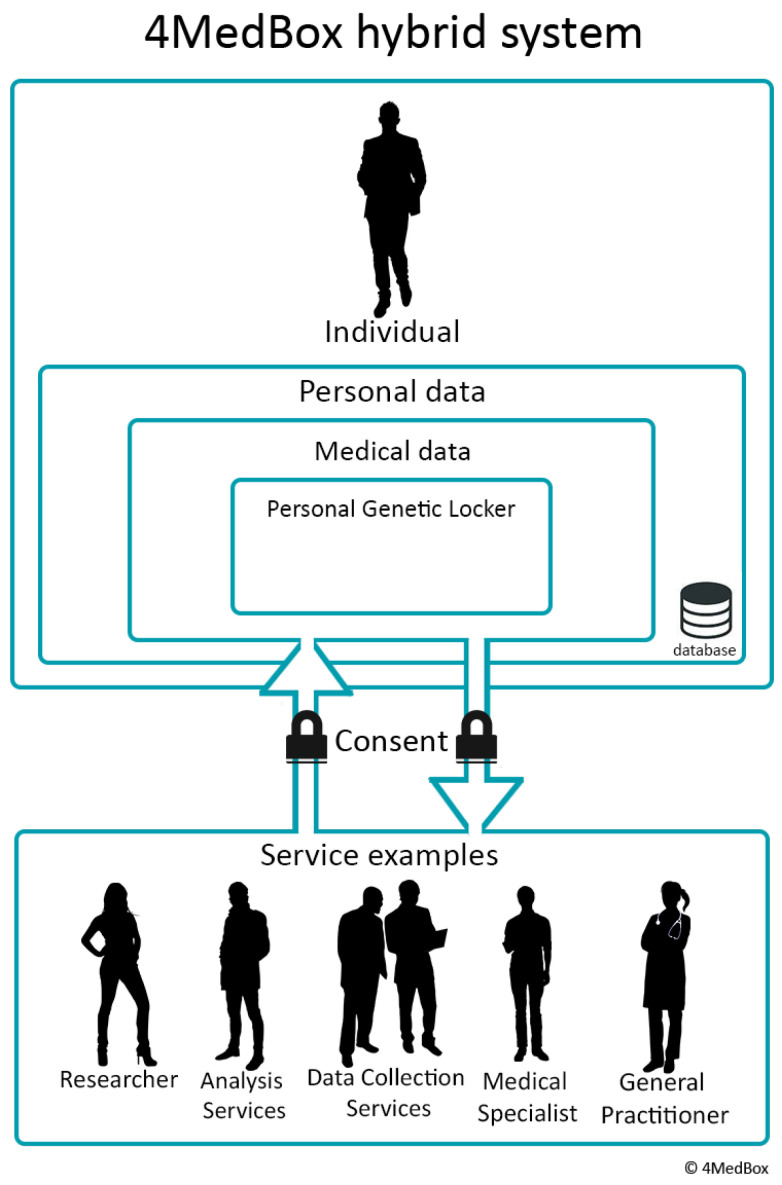
Schematic overview of the Personal Genetic Locker. The interaction between an individual and possible medical services. The image visualizes control over the data by the individual by giving consent. Reprinted from “Using personal genomic data within primary care: a bioinformatics approach to pharmacogenomics” [31]. Copyright 2020 by Overkleeft, R. Reprinted with permission.

**Figure 2 genes-14-00786-f002:**
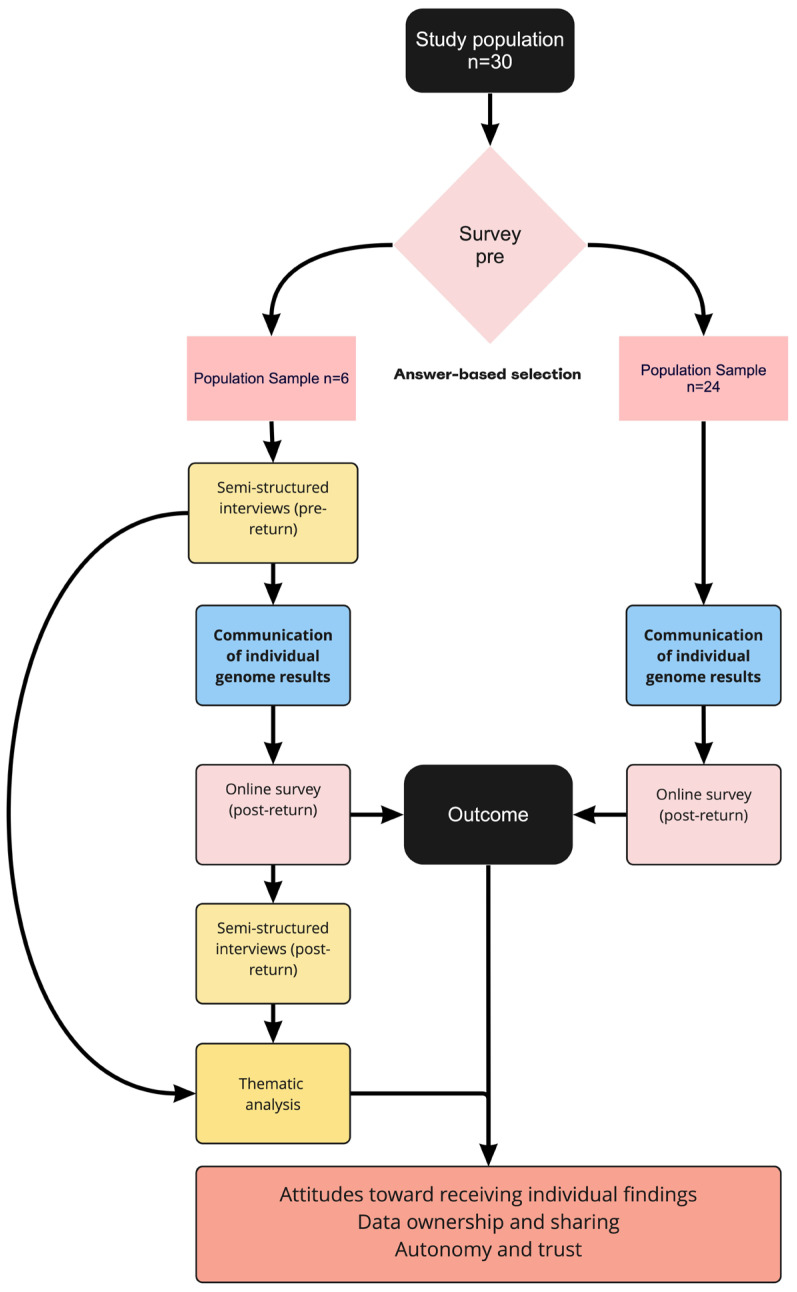
Overview of IAM Frontier survey and interview strategy (made in Miro).

**Figure 3 genes-14-00786-f003:**
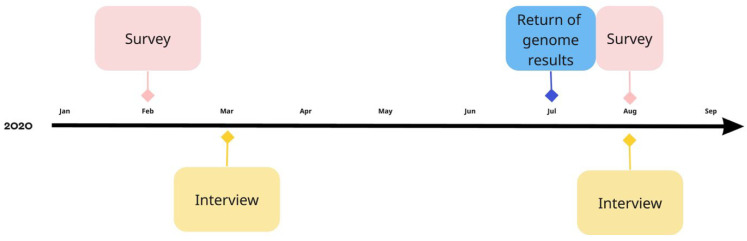
Timeline of survey questionnaires and interviews in the IAM Frontier study (made in Miro).

**Table 1 genes-14-00786-t001:** Demographic characteristics IAM Frontier and PGL participants.

	IAM Frontier			PGL Focus Groups		
		Survey Participants (*n* = 30)	Interview Participants (*n* = 6)		Healthcare Users (*n* = 6)	Healthcare Professionals (*n* = 4)
Gender						
	Male	15	3	Male	1	2
	Female	15	3	Female	5	2
Age, years						
	45–50	15	3	30–50	1	2
	51–58	15	3	51–70	5	2

Note. Values for categorical variables are given as counts. No participants presented as non-binary or gender-nonconforming. Different age categories were used in the studies.

**Table 2 genes-14-00786-t002:** Overview of participants in IAM Frontier interviews and PGL focus groups.

IAM Frontier			PGL Focus Groups		
	Gender	Age		Gender	Age
IAMF-P1	Male	51–58 years	PGL-P1	Female	30–50 years
IAMF-P2	Female	51–58 years	PGL-P2	Male	51–70 years
IAMF-P3	Female	51–58 years	PGL-P3	Female	51–70 years
IAMF-P4	Male	45–50 years	PGL-P4	Female	51–70 years
IAMF-P5	Male	45–50 years	PGL-P5	Female	51–70 years
IAMF-P6	Female	45–50 years	PGL-P6	Female	51–70 years
			PGL-P7	Female	30–50 years
			PGL-P8	Male	51–70 years
			PGL-P9	Female	30–50 years
			PGL-P10	Male	51–70 years

**Table 3 genes-14-00786-t003:** Meta-themes and subthemes generated from the IAM interviews and PGL focus groups.

Meta-Themes	IAM Healthy Participants	Healthcare Users	Healthcare Professionals
**Attitudes towards Receiving Genomic Information**			
Attitudes towards hypothetical genomic test results	Participants are interested in receiving sequencing results. Actionable results are welcomed as they enhance preventive capabilities and preparedness. (Temporarily) uncertain results might cause unnecessary worrying and impede decision making. Non-actionable results are favored as they can help to prepare and are valuable for end-of-life decisions. Personal and familial situation is highly influential for what one wants to know.	Being informed and being able to prepare for possible consequences as reasons for wanting to know test results. Personal decision, depends on personal and medical situation. It should always be the patient’s decision what to know and what not to know (i.e., right not to know).	There are regulations to prevent HCPs from knowing information that is unrelated to the diagnostic question. Diagnostic genetic test results are limited to answering a given diagnostic question. It should always be the patient’s decision what to know and what not to know (i.e., right not to know). Beforehand, patients sign a contract in which they indicate what type of results they want to be informed of and what not.
Attitudes after receiving individual genomic test results	Preferences are stable after undergoing sequencing. Disappointment and hope for more information.	-	-
**Attitudes towards ownership and sharing of genomic data**			
Between data ownership and data control	Genome data are highly personal. Research institutes have responsibilities to store sequencing data reliably, safely, and accessibly.	Patients should be the owner of their medical data. Support of a professional can be useful to explain the data.	Patients should be the owners of their medical data. Support of a professional is needed to explain the data. Support of a professional is needed to make sure data management is handled safely and responsibly.
Data sharing	Willingness to share with family members depends on nature of results and personal situation. Sharing with HCPs is favored and necessary for additional support. Participants are hesitant to share with commercial entities due to risk of abuse and lack of trust.	Certified HCPs are generally trusted in sharing genomic data. Data sharing increases the availability of knowledge, complete medical records, better and more efficient healthcare, and possibilities for tailoring treatment. Risk of data misuse, privacy concerns, and data leaks. Fear sharing medical records would cause HCPs judgement leading to false assumptions and prejudice. Two-way interaction: patients want something in return for sharing their data. They want to be informed, taken seriously, and work together.	Certified HCPs are trusted in sharing genomic data. Data sharing increases the availability of knowledge, complete medical records, better and more efficient healthcare, and possibilities for tailoring treatment. Risk of data misuse, privacy concerns, and data leaks. HCPs should have access only to what is relevant for the given context. Against sharing medical data with commercial companies and, to some degree, with insurance companies.
**Autonomy and trust**			
Autonomy as a central value	Autonomy in undergoing sequencing trumps societal benefits of genome sequencing. Autonomy in receiving results should always be respected. Individual autonomy is limited by moral obligations, vulnerability, and care relations.	It should be the patient’s decision what to know and what not to know. Patients want to be thoroughly informed, have access to their own data, and be involved in decision-making.	
The importance of trust in genomic medicine	Institutional trust is important for the implementation of genomic medicine. Trust in science is leveraged in light of uncertainties Interpersonal trust in HCPs is important and dependent on HCP competencies.	Trust is key in order to be willing to share data. Negative encounters with HCPs leads to distrust (e.g., being pushed from pillar to post, not being taking seriously, and not having access to all data).	Trust is key in order to be willing to share data.
Tension between patient protection and patient autonomy		Patients want to be in control.	HCPs want to protect and support.
**Applicability of Personal Health Data Spaces**			
Characteristics	-	(1) A secured environment, (2) owned and managed by the patient, (3) makes medical data available and accessible to patients, (4) saves and stores data in one place, and (5) allows patients to consent for their data being shared with HCPs.	(1) A secured environment, (2) owned and managed by the patient, (3) makes medical data available and accessible to patients, (4) saves and stores data in one place, and (5) allows patients to consent for their data being shared with HCPs.
Functionalities	-	Receiving “notifications” on relevant medical developments.	Receiving “notifications” on relevant medical developments. Supports in signaling genetic vulnerabilities. Adding interpretations or explanations to results. Possibility to share family history. Building “walls” to prevent unwanted or unsafe actions.
Type of data that should be stored in a PHDS	-	Regulations or infrastructure to prevent seeing unwanted test results. Living data that are responsive to medical developments.	Regulations or infrastructure to prevent seeing unwanted test results. Living data that are responsive to medical developments. Unsure what type of data would be most suitable (raw data, test results, whole genome).
Implementing a PHDS: barriers and facilitators	-		Barriers: security, technical and legal issues, limited health and digital literacy among patients, resistance in the public Facilitators: involving end-users in development, elaborate testing phase, cooperation, education, and training, and link to electronic health records.
Potential of a PHDS in healthcare	-	Yes. Reasons: In control of their own data, without any interference of or being dependent on HCPs, hospitals or institutions. Staying up-to-date on medical developments. Prevents data from getting lost.	Yes. Reasons: Facilitate the use of genomic information in clinical practice: it would improve the availability of data and would make the process more efficient.

Abbreviations. HCP: healthcare provider; PGL: Personal Genetic Locker; PHDS: Personal Health Data Space.

**Table 4 genes-14-00786-t004:** Respondents’ attitudes towards receiving hypothetical WGS results (pre, *n* = 30).

	Completely Disagree	Disagree	Neutral	Agree	Completely Agree
I am curious about my genetic predisposition to disease	1	1	2	6	20
I want to know as much as possible about my genome	0	1	0	11	18

Note. Pre: survey was administered before receiving WGS results. Values for categorical variables are given as counts. Abbreviations. WGS: whole genome sequencing.

**Table 5 genes-14-00786-t005:** Respondents’ attitudes towards receiving various categories of WGS findings (pre, *n* = 30).

	Yes	Don’t Know	No
Would you like to receive findings about:			
Probably no serious health risk	30	0	0
Life-threatening and preventable condition (actionable)	29	0	1
Serious conditions that are not life-threatening (non-actionable)	29	0	1
Non-preventable, life-threatening conditions (non-actionable)	24	5	1
Currently uncertain information (VUS)	15	10	5

Note. The survey was administered before receiving WGS results. Values for categorical variables are given as counts. Abbreviations. WGS: whole genome sequencing.

**Table 6 genes-14-00786-t006:** Respondents’ attitudes towards receiving various categories of WGS findings (post, *n* = 30).

	Yes	Don’t Know	No
Would you like to receive findings about:			
Probably no serious health risk	26	2	2
Life-threatening and preventable condition (actionable)	29	0	1
Non-preventable, life-threatening conditions (non-actionable)	19	7	4
Currently uncertain information (VUS)	16	5	9

Note. The survey was administered after receiving WGS results. Values for categorical variables are given as counts. Abbreviations. WGS: whole genome sequencing.

**Table 7 genes-14-00786-t007:** Respondents’ attitudes towards genome analysis after receiving individual WGS results (post, *n* = 30).

	Completely Disagree	Disagree	Neutral	Agree	Completely Agree
I am disappointed that the results of the genome analysis have not given me any more information.	2	5	6	14	3

Note. The survey was administered after receiving WGS results. Values for categorical variables are given as counts. Abbreviations. WGS: whole genome sequencing.

**Table 8 genes-14-00786-t008:** Respondents’ perceptions about sharing genomic results with different parties prior to receiving results (pre, *n* = 30).

	Completely Disagree	Disagree	Agree	Completely Agree	Don’t Know/No Answer
I am willing to share information about my genome with family	0	1	19	8	2
I am willing to share genomic information with researchers	0	0	4	24	2
I am willing to share genomic information with pharmaceutical companies	2	6	10	7	5

Note. The survey was administered before receiving WGS results. Values for categorical variables are given as counts. Abbreviations. WGS: whole genome sequencing.

**Table 9 genes-14-00786-t009:** Respondents on sharing genomic results with different parties after receiving results (post, *n* = 30).

	Yes	Don’t Know	No
Have you shared your genome results with blood relatives? (children, siblings, parents)	15	0	15
Have you shared your genome results with non-blood relatives? (e.g., partner, friends, …)	16	3	11

Note. The survey was administered after receiving WGS results. Values for categorical variables are given as counts. Abbreviations. WGS: whole genome sequencing.

**Table 10 genes-14-00786-t010:** Respondents’ preferred context for the implementation of genomic medicine (pre, *n* = 30).

	Completely Disagree	Disagree	Neutral	Agree	Completely Agree
In what context can genomic tests be performed according to you?					
Sold by a pharmacist	8	5	14	2	1
Only be performed with a doctor’s prescription	0	7	7	13	3
Sold through the internet	17	5	8	0	0
Offered by private companies	6	10	13	1	0
Only in the hospital	0	4	8	12	6

Note. The survey was administered before receiving WGS results. Values for categorical variables are given as counts. Abbreviations. WGS: whole genome sequencing.

**Table 11 genes-14-00786-t011:** Barriers to and facilitators of implementing a Personal Health Data Space in healthcare according to healthcare professionals.

Barriers	Facilitators
Security, technical, and legal issues	Involving end users in the development to improve the user-friendliness
Complex infrastructure	Elaborate testing phase
Difficulty in changing test possibilities and interpretations due to medical developments	Cooperation and long-term commitment (instead of multiple research projects)
Limited (health) literacy	Education on the benefits of PHDSs (users and HCPs)
Limited computer skills	Training HCPs to use PHDSs
Resistance, public opinion (privacy, safety)	Link PGL to existing electronic records
Costs: who is responsible?	
Access to genomic tests	

## Data Availability

The data presented in this study are available on request from the corresponding authors. The data are not publicly available due to privacy reasons.

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
