# Peer review of "Personal Genomes in Practice: Exploring Citizen and Healthcare Professionals’ Perspectives on Personalized Genomic Medicine and Personal Health Data Spaces Using a Mixed-Methods Design"

_genes, 2023, doi:10.3390/genes14040786_

Round 1

Reviewer 1 Report

The article is concerned with exploring personalized genomic medicine and Personal Health Data Spaces [PHDS] including concerns and benefits, from the perspectives of key stakeholders. These include healthy participants, healthcare users, and healthcare professionals. I very much enjoyed reading the article and believe that, bringing together these different perspectives, makes a valuable contribution to the field. The paper focuses on more general views of genomic medicine, as well as a specific solution - the Personal Genetic Locker - to some of the challenges that those living and working with genomic results face. The article is clear, well written and generally well structured. The introduction sets the scene by outlining the case for exploring stakeholder perspectives, drawing on examples from two different national contexts. The authors have employed a mixed methods approach, which is apt given the focus on stakeholder attitudes. A questionnaire survey is used to gain a sense of the breadth of experience, whilst qualitative methods (interviews and focus groups) are employed to explore depth. One specific value of this work is that participants were questioned hypothetically prior to having their genome sequenced and then again after they had participated in WGS. The outcomes of the qualitative and quantitative analysis are generally well synthesized in the results section. The discussion is very well presented and well-argued, and the conclusions are consistent with the analysis. The ethics statements and data availability statements are adequate.

There are three main areas that I feel would benefit from some refinement:

     (1)  The article, as it stands is 34 pages long and whilst qualitative and mixed methods papers warrant a generous word count to do justice to the depth and richness of the data, the results section could be more concise. It is easy for the reader to get lost in the detail and lose sight of the main messages the authors wish to convey. I don’t think it would be too onerous to tighten up the arguments made. For me, part of the issue concerns switching between findings from different types of data. I wonder if this could be integrated further, whilst also showing similarities and differences between views.

2 (2) Whilst the inclusion of a relatively small number of participants is not an issue in such work (and indeed in qualitative work the emphasis is depth of understanding) I do feel the paper would benefit from a stronger justification of the sample/those included. The authors state that they seek gain a sense of the diversity of experience and I would like to see further reflection on whether they felt this has been achieved (i.e. with a relatively small number of interviewees and focus group participants). I wondered if the inclusion criteria were too constraining.

3 (3) The paper also lacks detail regarding the approach to analysis, particularly the generation of themes, the process of interpretation and the synthesis of material generated using different methods. There is some discussion regarding the generation of codes but how the four meta-themes (i. attitudes toward receiving genomic information; ii. attitudes towards ownership and sharing of genomic data; iii. autonomy and trust; and iv. the applicability of the Personal Health Data Spaces) were established remains unclear. The process of determining/reconciling the meta-themes across the different elements of the analysis is also not apparent.

Specific comments

·       The term ‘general public’ is mentioned on multiple occasions (e.g. pages 2 and 3) but what is meant by general public is not explained. Would publics not be a more suitable term to highlight that this is not just one homogenous group?

·       Page 7 – Did all six participants take part in both interviews?

·       Page 9 (sections 2.4.2 and 2.4.3) - Why was different software used to aid the analysis of the interviews and the focus groups? Did this hinder the integration of findings?

·       Page 9 and 11 – the generation of meta-themes is mentioned (sections 2.4.4 and 3.2) but how were these generated? How were decisions made? Where these developed from the coding of the different data sets?

·       Page 9 (section 2.4.2) Please explain how the codes were reviewed by an independent researcher. What is an independent researcher? Does this fit with the Braun and Clarke approach to thematic analysis cited on page 9?

·       Did the focus groups comprise a mixture of those who had experienced genetic testing and those who had not? I would be interested in hearing more about the justification for mixing the groups (or not).

·       In Table 1 – why are the figures only reported for men? I assume the figures for women are meant to be inferred. This seems to be a very binary view of gender.

·       In the methods section (section 2.2.3) the authors state that they wish to include a diverse sample, but it would be helpful to see some reflection on how this was achieved with a small number of participants. I am not suggesting that the sample size ought to be bigger. There is tremendous value in small scale, in-depth qualitative work but it would be helpful to strengthen the justification, particularly for an audience likely to be more used to reading about large samples.

·       Table 3 (page 11) is very useful in highlighting how the themes connect across the samples but it isn’t clear how these cross-cutting themes were devised. How were they identified? To what extent did the way participants articulated their views shape code labelling?

·       In Table 8 I wonder who respondents might regard as researchers (‘I am willing to share genome information with researchers’)? (i.e. those working in Universities, industry etc.). The boundary between research and commerce might not be clear-cut.

·       The citations are comprehensive, up-to-date and appropriate with just one exception. It would be beneficial to explore Braun and Clarke’s more recent publications to help elaborate on the description of thematic analysis used e.g.

Braun, V., Clarke, V., Hayfield, N., Davey, L., and Jenkinson, E. (2023) Doing Reflexive Thematic Analysis, in Supporting Research in Counselling and Psychotherapy: Qualitative, Quantitative, and Mixed Methods Research, Springer International Publishing.

Reviewer 2 Report

Thank you for allowing me to review this manuscript. 

It offers relevant insights into the attitudes of healthcare users and professionals towards personalized genomic medicine. The manuscript is generally well-written, clear, well-structured, and has the potential to make an important contribution to the literature.

General remarks:

It would be valuable to explicitly, yet succinctly formulate the main research question that this paper addresses at the very start of the introduction. The introduction is rather long; briefly communicating the research question at the outset may help avoid that this focus is swamped before the reader arrives at the next sections.

I would appreciate a more thorough questioning in the discussion of the normative concepts that are employed in this paper (autonomy, trust, empowerment, ...). As an ethical matter this is somewhat underdeveloped. 

 Specific remarks:

p.2: Reference to ‘patient empowerment’: this is a buzz word that is commonly used by advocates of ‘personalized medicine’, and which mostly functions as an argument in its own terms. It has an overtly positive (emotive) connotation, though it is mostly unclear what it means. Skeptics may worry that without a more fine-grained understanding of this concept, it boils down to holding individuals responsible for their own health. It may be desirable to briefly reflect, or at least mention, some of these concerns. This becomes even more pressing in view of later invocations of ‘genuine empowerment’ (p.3). What is genuine empowerment? Did this come up as a concern that was shared by any of the participants?

Also: the assumption that having access to health information is empowering may be contested. Access to information is not per se synonymous with knowledge, which, in turn, is not synonymous with the ability to make the right choices based on that knowledge.

p.2: ‘Digital tools’: which digital tools?

p.3: “health data..”: please check the double ‘.’

p.21: references to trust: I would suggest to include a conception of what the authors understand as ‘trust’. In everyday speech, this is a seemingly clear concept, but upon reflection, and especially in the context of ethical thinking, it is less evident what this concept implies. First: what is ‘trust’ (do you here follow a standard account, something along the lines of: an expectation that a party will act in a certain way, combined with a normative expectation that this party should act that way?); second: how do you morally evaluate trust (is trust desirable, a good thing, a virtue, or is it a symptom of paternalism?). There is recent ethical literature that problematizes the relationship between trust, paternalism and empowerment also in the context of direct to consumer genetic testing and trends towards personalized medicine; it may be relevant to include this in your reflections. 

Reviewer 3 Report

This is an article about a fascinating topic: the plethora of new personal data spaces and how they interact with and may facilitate personalization of healthcare. I had been looking forward to the read. I found the article well-situated in an interesting literature and very well aware of the contemporary data landscape. These elements of the paper could easily deserve publication in a short and concise paper.

However, I think there are serious problems with the overall paper and its presentation of the empirical material as it stands now. It clearly embodies a lot of work and competence, but it sits uncomfortably between quantitative and qualitative traditions, between analysis and normative judgment, and between the types of conclusions it wishes to make and the material on which they are to be based. All in all, it comes across as a lack of reflectivity, which is probably more a question of format and style than the disposition of the authors. I will provide some examples of each of these elements below. They are meant as invitations for the authors to insist on the relevance of what they have done and the right to present the findings in a way that honors their qualitative work.

Tensions between quantitative and qualitative traditions:

It used to be a general rule that we would not report percentages when the population is below N=50. It is odd and strangely inflated to be so quantitative in the reporting and use SPSS for such small numbers. Conversely, the qualitative analysis is mostly limited to reporting what people said. There is hardly any analysis of tensions, contradictions, or inconsistencies in the positions of the informants – all the stuff that is at the heart of good qualitative research. The different subgroups are reported independently, rather than juxtaposed analytically. The quotes are rarely interpreted or explained. In this way the methodological quality is low by parameters from both sides – because it tries to live up to modes of presentation from both traditions. The inclusion of the coding scheme (table 3) is something we would see in student assignments, but hardly in a journal that takes qualitative research seriously. The “statistical analysis” on such small numbers really would not make it to a journal taking quantitative research seriously.

The inclusion and selection of people leaves a lot of questions open – it is clearly a biased group (selected according to their interests). It would not be a problem if a qualitative analysis reflected on the specificity and situatedness of the informants. However, when the paper makes conclusions about the “public” and “people’s” views, which is a quantitative form of generalization, then this material cannot carry this weight and the selection bias becomes problematic. The triangulation is claimed to substantiate the findings, but it is not explained and exemplified so that readers see what it did for the analysis. The inclusion criteria are stated, but not justified in the text. They are said to be aimed at a “balance”, but with e.g. just six people representativity is not relevant (it is a quantitative demand). Rather, you want maximum variance.

Tensions between analysis and normative judgment:

The article claims to present an analysis, but it also promotes a number of more normative judgments and recommendations, not all of which are clearly supported by explication of the values that inform them (how to move from is to ought), and the evidence that they are feasible to pursue. I can only provide a couple of examples but I encourage the authors to rethink their normative claims in general. The introduction asserts that the challenges of rising costs can be countered with personalization. Most studies suggests that personalization in the form of genomic medicine is a driver of rising costs. It is not a way to save money – it is only something you see as claims in policy reports, not in serious academic literature dealing with the economics of personalized medicine. Another example: Do references 8, and 12-14, really support the claim about general attitudes in the public? Which publics where?

Tensions between the types of conclusions it wishes to make and the material on which they are to be based:

The conclusion states that “This study aimed to evaluate the applicability of genomic medicine and PDS initiatives”. However, this type of conclusion cannot be reached with this type of material.  The authors conclude that “Our findings support earlier reports signaling the need to implement genomic medicine in current healthcare systems,” but this type of normative goal seems to be what motivated the study rather than a finding. One could argue that your findings point in the opposite direction, as most people – in a highly select pro-tech subgroup of the wider population – were disappointed about the lack of usability of genomic data.

Lack of reflectivity:

I really like the comments on the deficiencies of EHDS. However, when PHDSs are presented as a solution, it comes across as somewhat unsubstantiated. If they are intermediaries, how will they solve the EHDS problems? I also like the discussion of views of ownership, but I see few reflections on how this topic seems to have been introduced by the authors and what this did in terms of framing their responses. I also like that you discuss autonomy AND how genetic information is shared in families, but I see no clear discussion of the tensions and questions it raises to think of it in contradictory ways as both private and shared. Similarly, with the conclusions – do they really follow from your findings or are they what you wanted healthcare systems to do all along?

Please note also that not all references are full and adequately cited.  

I hope you can use these comments to place yourself more firmly in a tradition that allows you to unfold and describe the new findings in the material you’ve gathered.

Round 2

Reviewer 2 Report

Thank you for the changes you included in your revised manuscript. All my previous remarks have been dealt with in an accurate manner.